# IL-1 Generated by Oral Squamous Cell Carcinoma Stimulates Tumor-Induced and RANKL-Induced Osteoclastogenesis: A Possible Mechanism of Bone Resorption Induced by the Infiltration of Oral Squamous Cell Carcinoma

**DOI:** 10.3390/ijms24010688

**Published:** 2022-12-30

**Authors:** Yuki Fukawa, Kou Kayamori, Maiko Tsuchiya, Tohru Ikeda

**Affiliations:** Department of Oral Pathology, Graduate School of Medical and Dental Sciences, Tokyo Medical and Dental University, 1-5-45 Yushima, Bunkyo-ku, Tokyo 113-8549, Japan

**Keywords:** osteoclastogenesis, oral squamous cell carcinoma, IL-1, osteoclast precursor cells, cannabidiol

## Abstract

We previously observed a novel osteoclastogenesis system that is induced by oral squamous cell carcinoma (OSCC) cells, which target osteoclast precursor cells (OPC) without upregulation of the master transcriptional factor of osteoclastogenesis, NFATc1. Here, we analyzed inflammatory cytokines that were preferentially expressed in one of the osteoclastogenic OSCC cell lines, namely NEM, compared with the subclone that had lost its osteoclastogenic properties. Based on a gene expression microarray and a protein array analyses, IL-1, IL-6, IL-8, and CXCL1 were chosen as candidates responsible for tumor-induced osteoclastogenesis. From the results of the in vitro osteoclastogenesis assay using OPCs cultured with OSCC cells or their culture supernatants, IL-1 was selected as a stimulator of both OSCC-induced and RANKL-induced osteoclastogenesis. The IL-1 receptor antagonist significantly attenuated osteoclastogenesis induced by NEM cells. The stimulatory effects of IL-1 for OSCC-induced and RANKL-induced osteoclastogenesis were effectively attenuated with cannabidiol and denosumab, respectively. These results suggest that IL-1 secreted from OSCC cells stimulates not only tumor-induced osteoclastogenesis targeting OPCs but also physiological RANKL-induced osteoclastogenesis, and this may be the biological mechanism of bone resorption induced by the infiltration of OSCC. These results also suggest that IL-1 inhibitors are candidates for therapeutic agents against bone resorption induced by OSCC.

## 1. Introduction

Bone metabolism is regulated by the balance of bone resorption, mediated by osteoclasts and bone formation induced by osteoblasts [1]. The efficient regulation of bone metabolism was confirmed by clarifying the mechanism of the RANKL/RANK system [2]. The outline of the major osteoclastogenesis pathway is an interaction of RANKL, which is expressed in osteoblasts and osteocytes at the receptor RANK, found in macrophage-derived osteoclast progenitor cells maintained with macrophage colony-stimulating factor (M-CSF) [2,3]. The signal from activated RANK induces NFATc1, the master transcriptional factor of osteoclastogenesis, by way of the TRAF6 pathway [1]. Additionally, Wnt signaling is associated with bone metabolism, including osteoclastogenesis [4]. Bone metastasis of cancer cells greatly worsens a patient’s quality of life and prognosis [5]. Prostatic cancer is highly metastatic to the bone tissue, but clinically exhibits osteoplastic reactions in most cases [6]. In contrast, most bone metastases induced by other cancers reveal osteolytic reactions caused by the induction and activation of osteoclasts [7,8]. The osteolytic reaction is also observed in direct tumor invasion into the bone, which is frequent in oral squamous cell carcinoma (OSCC) cases [9]. Hence, clarifying the mechanism of tumor-induced osteoclastogenesis is essential to develop therapeutic agents against tumor-induced bone resorption.

Previous studies of tumor-induced osteoclastogenesis can be divided into those focusing on the generation of RANKL by tumor cells, the generation of effectors of RANKL-induced osteoclastogenesis, and the generation of TNF-α, which alone induces osteoclasts [10,11,12,13,14,15,16,17,18]. Reports of the expression of RANKL by tumor cells are limited, and RANKL is not widely expressed in head and neck carcinomas [19]. Considering the high frequency of bone invasion of head and neck carcinomas, some inconsistencies might be present. Another hypothesis is that tumor cells stimulate the stromal cells to express RANKL [10]. However, information to support this kind of hypothesis is limited, and the accumulation of supportive data is needed. TNF-α is expressed in some kinds of carcinomas including head and neck carcinomas [20]. However, TNF-α was not considered the main osteoclastogenic pathway, and the biological significance was considered to be associated with some inflammatory osteolytic reactions [3,21]. In addition, inhibitors of TNF-α, which are applied in rheumatoid arthritis, have not been reported to successfully attenuate tumor-induced bone resorption.

Several kinds of inflammatory cytokines, e.g., IL-1α, IL1-β, and IL-6, are listed as major stimulators of RANKL-induced osteoclastogenesis [21,22]. IL-1 (including both IL-1α and IL1-β) and IL-6 stimulate RANKL-induced osteoclastogenesis under inflammatory conditions, but they are also associated with tumor-induced osteoclastogenesis in vitro and in vivo [10,23]. IL-1, which is expressed in head and neck carcinomas, was suggested to play a pivotal role in bone resorption induced by tumor invasion or metastasis [9,24,25]. Moreover, IL-1 expresses osteoclastogenic activity in macrophages with artificial overexpression of IL-1 receptor 1 in the presence of M-CSF [26]. However, the expression of RANKL in head and neck carcinomas is generally low, and the self-stimulation of RANKL expression by IL-1 would not be the major mechanism of bone resorption induced by head and neck carcinomas. In addition, the stimulation of stromal cells to express RANKL with tumor-derived IL-1 lacks evidence, so far. Hence, the biological significance of IL-1 in bone resorption induced by head and neck carcinomas including OSCC remains unclear.

Previously, we found that human OSCC cell lines possessed potent osteoclastogenic properties when cocultured with macrophages exposed to RANKL for 24 h in the presence of M-CSF, which are defined as osteoclast precursor cells (OPC) [27]. Importantly, OSCC cells failed to induce osteoclasts from RANKL untreated macrophages in the presence of M-CSF, which are defined as osteoclast progenitor cells. OPCs are cells that are dedicated to osteoclastic differentiation. However, they do not express any osteoclast functions and are unable to differentiate into osteoclasts without further stimulation by RANKL. Osteoclastogenesis induced by OSCC cells targeting OPCs was resistant to RANKL inhibitory agents, such as osteoprotegerin (OPG) and denosumab, in contrast to RANKL-induced osteoclastogenesis [27,28]. Furthermore, osteoclastogenesis induced by OSCC cells did not correlate with the upregulation of NFATc1 expression, and tumor-induced osteoclastogenesis, which targets OPCs, was caused by unknown mechanisms. Thereafter, we found that osteoclastogenesis induced by OSCC cells was partly caused by secreted extracellular microvesicles, and cannabidiol (CBD), one of the cannabinoids that does not express psychotomimetic functions, effectively attenuated osteoclastogenesis induced by OSCC cells and did not affect RANKL-induced osteoclastogenesis [28]. In this study, we aimed to clarify the unknown factors responsible for osteoclastogenesis induced by OSCC cells using a subclone of OSCC cells which did not have osteoclastogenic properties, selected IL-1 as a stimulator of both tumor-induced and RANKL-induced osteoclastogenesis, and examined the biological significance of IL-1 in bone resorption induced by OSCC. The results of this study will provide a novel concept regarding the mechanism of bone resorption induced by OSCC and will lead to the development of therapeutic agents against bone resorption induced by OSCC.

## 2. Results

### 2.1. Induction of Osteoclasts from OPCs by OSCC Cell Lines

We analyzed the induction of osteoclasts from OPCs cocultured with different human-derived OSCC cell lines. All eleven OSCC cell lines, NEM, 3A, Ca, HSC-2, HSC-3, HSC-6, NA, NU, OMI, SH, and Toh, revealed osteoclastogenesis when cocultured with OPCs (Appendix A). Osteoclastogenesis induced by these cell lines was resistant to OPG (Appendix A). By screening the subclones derived from NEM cells, two subclones were selected and used for the following study, namely NEM-F, which possessed osteoclastogenic properties, and NEM-K, which did not possess osteoclastogenic properties. The osteoclastogenic properties induced by the coculture of NEM-F and OPCs were comparable to those induced by the treatment of 100 ng/mL of RANKL, and the culture supernatant of NEM-F cells induced osteoclasts from OPCs (Figure 1a,b). Notably, OPCs without further stimulation by RANKL failed to differentiate into osteoclasts (Figure 1a,b). Induction of osteoclasts using the culture supernatant of NEM-F confirmed the presence of certain factors or vacuoles secreted from the cancer cells, which participated in osteoclastogenesis. Conversely, NEM-K did not exhibit osteoclastogenic properties (Figure 1a,b). Comprehensive analysis of the expression genes between NEM-F and NEM-K revealed preferential expression of several inflammatory cytokines, which have been associated with osteoclastogenesis. Considering the results of gene expression microarray analysis in our previous study, we selected IL-1α, IL-1β, CXCL1, and IL-8 as candidates that may be responsible for tumor-induced osteoclastogenesis (Figure 1c, red bars). These four factors were also substantiated by the protein array analysis (Appendix A). In addition, IL-6 was added to the candidates because it was preferentially expressed in NEM-F cells compared with NEM-K cells in this study, although the expression was not different in our previous gene expression microarray analysis.

### 2.2. Evaluation of Induction or Stimulation of Osteoclastogenesis with IL-1α, IL-1β, IL-6, IL-8, and CXCL1

The osteoclastogenic activity of selected inflammatory cytokines, IL-1α, IL-1β, IL-8, and CXCL1, was analyzed by adding these four factors into cultures of OPCs in the growth medium or coculture with NEM-K. IL-1α, IL-1β, IL-8, and CXCL1 did not induce osteoclasts from OPCs, and these four factors did not have osteoclastogenic effects on OPCs. In contrast, these four factors significantly stimulated osteoclastogenesis induced by coculture with NEM-F (Figure 2a,b). The stimulatory effect of each factor was also evaluated by supplementation into OPCs cocultured with NEM-F. As shown in Figure 2c, IL-1α and IL-1β significantly stimulated osteoclastogenesis induced by coculture with NEM-F, but IL-8 and CXCL1 did not. The stimulatory potency of IL-1α and IL-1β was comparable, and no significant difference was detected between them (Figure 2c). Then, using IL-1α, stimulation of RANKL-induced osteoclastogenesis was also evaluated. As detected in tumor-induced osteoclastogenesis, IL-1α significantly stimulated RANKL-induced osteoclastogenesis (Figure 2d,e). Denosumab completely inhibited RANKL-induced osteoclastogenesis, irrespective of supplementation with IL-1α. In contrast, denosumab did not affect osteoclastogenesis induced by coculture with NEM-F, irrespective of supplementation with IL-1α (Figure 2d,e). In addition to these four factors, we also analyzed the influence of IL-6 on tumor-induced osteoclastogenesis. IL-6 did not stimulate osteoclastogenesis induced by coculture with NEM-F cells (Appendix A).

### 2.3. The Effect of Inhibitors for IL-1α and IL-1β on the Tumor-Induced Osteoclastogenesis

To evaluate the stimulatory effect of IL-1 on tumor-induced osteoclastogenesis, IL-1 receptor antagonist (IL-1RA), an inhibitor for IL-1α and IL-1β or an antagonistic antibody for IL-1R1, a signaling receptor for both IL-1α and IL-1β, was applied to OPCs cultured with the culture supernatant of NEM-F. IL-1RA and anti-IL-1R1 antagonistic antibodies effectively attenuated osteoclastogenesis induced by the culture supernatant of NEM-F (Figure 3a,b).

The expression of the master transcriptional factor for osteoclastogenesis, NFATc1, was analyzed by quantitative PCR analysis, and no significant increase in the expression of NFATc1 was detected in osteoclastogenesis induced by the culture supernatant of NEM-F, irrespective of supplementation with IL-1α or IL-1RA, in contrast with the expression in RANKL-induced osteoclastogenesis (Figure 3c). Although the expression profile of NFATc1 was different, an osteolytic function was observed in osteoclasts induced by coculture with NEM-F, similar to RANKL-induced osteoclasts (Figure 3d,e). Osteolytic function in osteoclasts induced by coculture with NEM-F was significantly promoted by IL-1α and significantly inhibited by IL-1RA compared with cultures that were not supplemented (Figure 3d,e).

### 2.4. The Effect of CBD and IL-1RA on Tumor-Induced and RANKL-Induced Osteoclastogenesis

CBD effectively attenuated osteoclastogenesis induced by the culture supernatant of NEM-F, but it was ineffective in attenuating RANKL-induced osteoclastogenesis. Conversely, denosumab completely inhibited RANKL-induced osteoclastogenesis, but it was ineffective in attenuating osteoclastogenesis induced by the culture supernatant of NEM-F (Figure 4a,b). The inhibitory effect of CBD on osteoclastogenesis induced by OSCC cells was also confirmed in the coculture of OSCC cells derived from 3A, OMI, and NA with OPCs (Appendix A). IL-1RA also significantly attenuated osteoclastogenesis induced by the culture supernatant of NEM-F, but the inhibitory effect of CBD on osteoclastogenesis was more potent compared with that of IL-1RA. However, the combination of CBD and IL-1RA did not improve the inhibition of osteoclastogenesis induced by the culture supernatant of NEM-F, and the combination was ineffective in attenuating RANKL-induced osteoclastogenesis (Figure 4a,b).

### 2.5. The Effects of Agonistic Interactors for GPR55 on Tumor-Induced Osteoclastogenesis

The effects of agonistic interactors for GPR55, such as LPI, O-1602, and ABN-CBD, on osteoclastogenesis induced by the culture supernatant of NEM-F cells were analyzed. O-1602 and ABN-CBD significantly increased the number of osteoclasts induced by the culture supernatant of NEM-F cells, but LPI did not. Significant attenuation of tumor-induced osteoclastogenesis was induced by one of the GPR55 antagonists, CBD (Figure 5).

## 3. Discussion

We have shown that OSCC cell lines express potent osteoclastogenic properties when cocultured with OPCs, which are derived from mouse bone marrow macrophages treated with M-CSF and RANKL for 24 h, as defined by Mizoguchi et al. [27,29]. In contrast, OSCC cell lines revealed poor osteoclastogenic activity when cocultured with osteoclast progenitor cells, which were derived from mouse bone marrow macrophages treated with M-CSF alone [27]. Furthermore, tumor-induced osteoclastogenesis targeting OPCs was observed in all the OSCC cell lines except for NEM-K, as shown in this study. These results strongly suggest that tumor-induced osteoclastogenesis targeting OPCs can be relatively common for OSCC cells. Histopathological findings of OSCC cases with infiltration into the bone reveal the interposition of fibrous stromal tissue between cancer cells and bone surfaces. Culture supernatants of OSCC cells revealed osteoclastogenesis when cultured with OPCs, and we identified the cause as secreted vesicles or certain soluble factors. In the previous study, we showed that extracellular microvesicles secreted from OSCC cells that had osteoclastogenic properties induced osteoclastogenesis when supplemented with OPC cultures [28]. Notably, the depletion of extracellular microvesicles from the culture supernatant significantly decreased the osteoclastogenic activity, but not completely [28]. Hence, to search for other candidates, we analyzed soluble factors secreted by OSCC cells.

Among numerous soluble factors, some inflammatory cytokines have been shown to stimulate osteoclastogenesis [21]. TNF-α is ubiquitously expressed in many types of cancer cells, including OSCC cells, and it was shown that TNF-α alone induced osteoclastogenic activity [14,20,30,31,32,33,34]. Hence, it was suggested that TNF-α participated in pathological bone resorption. In this study, protein array analysis revealed that TNF-α was expressed in osteoclastogenic and non-osteoclastogenic OSCC cells. In addition, we showed in a previous study that infliximab, one of the inhibitory agents for TNF-α activity, was ineffective in attenuating tumor-induced osteoclastogenesis [27]. Therefore, TNF-α was excluded from the causative factors for tumor-induced osteoclastogenesis. Considering the results of the gene expression and protein arrays in this study, along with the gene expression microarrays in our previous study, IL-1α, IL-1β, IL-8, and CXCL1 were selected as candidates for OSCC-induced osteoclastogenesis. IL-6 was also analyzed, which was preferentially expressed in NEM-F compared with NEM-K but exhibited no obvious differences between osteoclastogenic and non-osteoclastogenic OSCC cells in a gene expression microarray in our previous study. Among the candidate factors, IL-1α and IL-1β were suggested to have osteoclastogenic activity and were thought to be strong participants in tumor-induced osteoclastogenesis. Although none of these selected factors, alone or in combination, induced osteoclasts from OPCs, IL-1α and IL-1β stimulated tumor-induced and RANKL-induced osteoclastogenesis. IL-1 is one of the potent stimulators of osteoclastogenesis and osteoclast activities. The mechanism for this stimulation was suggested to be an increase in the expression of RANKL, which leads to upregulated expression of NFATc1, the master transcriptional factors for osteoclastogenesis [35,36,37], although a similar function was suggested to be expressed by IL-6 and IL-8 [21]. Additionally, IL-1 directly stimulated osteoclastogenesis in the presence of a minimal dose of RANKL [36]. In the present study, IL-6 and IL-8 were preferentially expressed in NEM-F compared with the non-osteoclastogenic subclone, NEM-K. However, there was no effect on tumor-induced osteoclastogenesis.

Both IL-1α and IL-1β stimulated tumor-induced osteoclastogenesis. Furthermore, IL-1α alone may exhibit osteoclastogenic activity when supplemented with cultures of macrophages that have an overexpression of IL-1R1, and it was reported to be independent of NFATc1 upregulation [26]. These results suggest that a potent signal of IL-1R1 induces osteoclasts by an independent pathway of RANKL-induced osteoclastogenesis. We showed that the number of osteoclasts induced by OSCC cells from OPCs did not correspond with the upregulation of NFATc1 expression. Thus, osteoclastogenesis induced by OSCC cells from OPCs was an alternate pathway of RANKL-induced osteoclastogenesis, and there may be multiple unknown pathways of osteoclastogenesis after the induction of OPCs with brief or weak exposure to RANKL. Notably, IL-1α stimulated tumor-induced osteoclastogenesis and RANKL-induced osteoclastogenesis. Moreover, with or without supplementation of IL-1α, tumor-induced osteoclastogenesis was sensitive to CBD and resistant to denosumab, whereas RANKL-induced osteoclastogenesis was resistant to CBD and sensitive to denosumab. These results suggest that stimulatory effects of IL-1α for osteoclastogenesis were expressed by acting on cells in more mature stages than OPCs, which appeared in both OSCC-induced and RANKL-induced osteoclastogenesis. Another possibility is that IL-1α acts on OPCs and independently alters the signaling pathways of tumor-induced and RANKL-induced osteoclastogenesis.

Herein, tumor-induced osteoclastogenesis targeting OPCs was observed in most of the analyzed OSCC cell lines. Hence, developing inhibitors of this osteoclastogenesis pathway may lead to novel therapeutic agents for bone resorption caused by OSCC infiltration. We also clarified that CBD effectively attenuated tumor-induced osteoclastogenesis, and thus, detailed studies of cannabinoids and their receptors are required [28]. Endocannabinoids have been shown to affect bone metabolism, and several cannabinoid derivatives were reported to stimulate bone regeneration [38]. Although cannabinoids are recognized as one of the modulators of bone metabolism, the mechanism remains unclear. Some reports have suggested that cannabinoids stimulate osteogenesis through the CB2 receptors [39,40,41], and cannabinoids were reported to affect osteoclast activity and osteoclastogenesis [39,42]. However, compared with osteogenesis, information regarding the effects of cannabinoids on osteoclasts is more limited. While the biological functions of CBD remain unclear, the lack of psychotomimetic effects enables us to apply it therapeutically [43]. A previous study suggested that CBD interfered with osteoclastogenesis by binding with one of the interactors, GPR55 [44]. In this study, we applied GPR55 agonists, such as LPI, O-1602, and ABN-CBD, to cultures of OPCs using a culture supernatant of NEM-F cells. O-1602 and ABN-CBD significantly increased the number of osteoclasts, which was in contrast with the inhibitory effect of CBD, one of the antagonists for GPR55. These results suggest that OSCC-induced osteoclastogenesis was associated with GPR55 signaling; however, future studies of the relationship between the inhibitory mechanism of CBD and GPR55 signaling in tumor-induced osteoclastogenesis are necessary.

Moreover, IL-1 is expressed in head and neck carcinomas [24,25,32]. In the current study, we showed that IL-1RA, an inhibitor for IL-1α and IL-1β, significantly attenuated osteoclastogenesis induced by NEM-F cells. These results suggest that IL-1 generated by OSCC cells participated in tumor-induced osteoclastogenesis. In addition, we confirmed the stimulatory effect of IL-1α on RANKL-induced osteoclastogenesis. Overall, IL-1 secreted from OSCC cells may spread by diffusion, act on OPCs and osteoclast progenitor cells near the bone surface and stimulate not only tumor-induced osteoclastogenesis targeting OPCs but also physiological RANKL-induced osteoclastogenesis on the bone surface. Hence, IL-1 inhibitors may effectively attenuate bone resorption caused by OSCC infiltration by suppressing the stimulatory effect of IL-1 on tumor-induced and conventional RANKL-induced osteoclastogenesis (Figure 6). In this study, there are no in vivo data, and further animal experiments are needed to confirm our hypothesis. In addition, IL-1RA, which has been applied clinically for the treatment of rheumatoid arthritis in some countries, may be one of the potential therapeutic agents against OSCC-induced bone resorption. Considering that most of OSCC cell lines possess osteoclastogenic properties from OPCs, the results of this study provide valuable information for developing new therapeutic agents against bone resorption induced by OSCC, contributing to the improvement of prognoses and quality of life among OSCC patients.

## 4. Materials and Methods

### 4.1. Culture of OSCC Cells

Human OSCC cell lines, such as NEM, 3A, Ca, HSC-2, HSC-3, HSC-6, NA, NU, OMI, SH, and Toh, were cultured in Dulbecco’s Modified Eagle’s Medium (DMEM) (Sigma-Aldrich, St. Louis, MO, USA) with 10% fetal bovine serum (HyClone, Logan, UT, USA) and 1% penicillin-streptomycin (Sigma-Aldrich). These cell lines were kindly provided by Professor Emeritus Nobuo Tsuchida, from the Tokyo Medical and Dental University. These cells were maintained in the growth medium until just before performing the osteoclastogenesis assay by means of cocultures with OPCs or the collection of culture supernatants provided for the osteoclastogenesis assay. Subclones of NEM cells were isolated using the method of dilution culturing. Subclones grown from single cells were selected by evaluating the osteoclastogenic properties in the presence of OPCs. In some experiments, we used NIH3T3 cells with overexpression of human RANKL and mouse M-CSF, which were named 1+1+M1 cells for RANKL-induced osteoclastogenesis by coculturing with OPCs [45].

### 4.2. Osteoclastogenesis Assay Using OPCs

Mouse bone marrow macrophages were prepared as described previously, with minor modifications. The bone marrow tissue was flushed out from the femurs and tibiae obtained from 5-week-old ddY mice (Sankyo Lab Service, Shizuoka, Japan). The bone marrow tissue was dissociated by pipetting and centrifuged with Lymphoprep^TM^ (Axis Shield PoC AS, Oslo, Norway) following the manufacturer’s instructions to isolate macrophages. The isolated macrophages were cultured in the alpha modified Minimum Essential Medium (MEM-α) (Sigma-Aldrich, St. Louis, MO, USA) containing 10% fetal bovine serum (HyClone), 1% penicillin-streptomycin-glutamine (Sigma-Aldrich), and 30 ng/mL of mouse M-CSF (416-ML, R&D Systems, Minneapolis, MN, USA) for 3 days. The expanded macrophages were further cultured in the medium supplemented with 100 ng/mL of recombinant human RANKL (Oriental Yeast, Tokyo, Japan) for 24 h to generate OPCs, as described previously [27,28]. The generated OPCs were subjected to osteoclastogenesis assays by coculture with OSCC cells or using a culture supernatant of OSCC cells, which was collected from cells maintained in DMEM and cultured in the MEM-α for 2 days. Animal experiments were performed following the Guidelines for Animal Experimentation of Tokyo Medical and Dental University, with the official approval of the committee (approval no. A2021-113C2).

In the coculture osteoclastogenesis assay, 3 × 10^3^ OSCC cells and 3 × 10^4^ OPCs were mixed and cultured for 5 days in MEM-α supplemented with 30 ng/mL of M-CSF in each well of a 48-well plate. In osteoclastogenesis using a culture supernatant, 3 × 10^4^ OPCs were cultured in MEM-α containing 60% of the fresh medium and 40% of the culture supernatants of OSCC cells with 30 ng/mL of M-CSF for 5 days in each well of a 48-well plate. Conventional osteoclastogenesis was induced by culturing 3 × 10^4^ OPCs in the medium supplemented with 30 ng/mL of M-CSF and 100 ng/mL of recombinant human RANKL for 5 days in each well of a 48-well plate. Osteoclastogenesis was evaluated after 5 days of culture and staining for tartrate-resistant acid phosphatase (TRAP) activity using the method described previously [27,28]. TRAP-positive cells with three or more nuclei were considered osteoclasts.

In some of these cultures, 100 μg/mL of denosumab (Ranmark, Daiichi-Sankyo Co., Ltd., Tokyo, Japan), 300 ng/mL of osteoprotegerin (OPG) (Peprotech, Rock Hill, NJ, USA), 100 ng/mL of IL-1α (098-06801, FUJIFILM Wako Pure Chemical, Osaka, Japan), 100 ng/mL of IL-1β (090-06121, FUJIFILM Wako Pure Chemical), 100 ng/mL of IL-6 (206-IL, R&D Systems), 100 ng/mL of IL-8 (574204, BioLegend, San Diego, CA, USA), 100 ng/mL of CXCL1 (275-GR, R&D Systems), 10 μg/mL of IL-1 receptor antagonist (IL-1RA) (093-05991, FUJIFILM Wako Pure Chemical) or 5 μg/mL of anti-IL-1 receptor 1 (IL-1R1) antagonistic antibody (PA5-47937, Invitrogen, Carlsbad, CA, USA) was further added to analyze the effects on osteoclastogenesis. In the experiment using GPR55 interactors, 5 μM of CBD (C6395, Sigma-Aldrich), 1 μM of L-α-Lysophosphatidylinositol sodium salt from soybean (LPI) (62966, Sigma-Aldrich), 1 μM of O-1602 (ab120407, Abcam, Cambridge, UK), or 5 μM of abnormal-cannabidiol (ABN-CBD) (sc-203488B, Santa Cruz Biotechnology, Santa Cruz, CA, USA) was further added to the culture using a culture supernatant of OSCC cells. Osteoclastogenesis was evaluated after 5 days of culture and staining for TRAP activity.

### 4.3. Pit Formation Assay Using OPCs

A total of 3 × 10^3^ OSCC cells and 3 × 10^4^ OPCs were mixed and cultured for 6 days in MEM-α supplemented with 30 ng/mL of M-CSF on the Bone Resorption Assay Plate (BRA-48P, PG Research, Tokyo, Japan). In some of these cocultures, 100 ng/mL of IL-1α or 10 μg/mL of IL-1RA was further added to analyze the effects on the osteolytic function, which was evaluated by the number of pits.

### 4.4. Gene Expression Microarray Analysis and Protein Array Analysis

Total RNA was extracted from NEM-F and NEM-K cells under sub-confluent conditions using a NucleoSpin RNA Kit (Macherey-Nagel, Duren, Germany). After qualification and quantification of the extracted RNA, gene expression microarray analysis using the Clariom^TM^ D Assay for human samples (Affymetrix, Santa Clara, CA, USA) was performed by Filgen Inc., Nagoya, Japan. Culture supernatants of NEM-F and NEM-K were prepared for the protein array following the manufacturer’s instructions. After purification, qualification, and quantification of the NEM-F and NEM-K culture supernatants, protein array analysis using RayBio^®^ Label-Based (L-Series) Human Antibody Array 1000 (RayBiotech, Inc., Peachtree Corners, GA, USA) was performed by Filgen Inc., Nagoya, Japan.

### 4.5. Quantitative RT-PCR

Total RNA was extracted after 5 days of culture using a NucleoSpin RNA Kit (Macherey-Nagel), reverse-transcribed into cDNA according to the method described previously, and subjected to quantitative real-time PCR using a Light Cycler Nano (Roche Diagnostics, Base, Switzerland) with the FastStart Essential DNA Master Mix (Roche Diagnostics, Penzburg, Germany) [27,28]. The primers used to analyze the expression of mouse NFATc1 and mouse GAPDH had the following sequences: mNFATc1, 5′-TGCTCCTCCTCCTGCTGCTC-3′ (forward) and 5′-CGTCTTCCACCTCCACGTCG-3′ (reverse); mGAPDH, 5′-CATGGCCTTCCGTGTTCCTA-3′ (forward) and 5′-GCGGCACGTCAGATCCA-3′ (reverse). The relative expression level of each mRNA was calculated using the comparative CT method with GAPDH as an internal control, and each experiment was repeated at least three times.

### 4.6. Statistical Analysis

All results were recorded as the mean ± SD of independent replicates. The difference between the two groups was analyzed using Student’s *t*-test. The difference among three or more groups was analyzed using a one-way analysis of variance and Tukey–Kramer or Dunnett’s post hoc multiple comparisons test. Results with *p*-values < 0.05 were considered significant.

## 5. Conclusions

We analyzed the biological significance of IL-1 in tumor-induced osteoclastogenesis and clarified that IL-1 stimulated osteoclastogenesis induced by OSCC cells from OPCs without affecting the expression of NFATc1. IL-1 also stimulated RANKL-induced osteoclastogenesis. With or without IL-1 supplementation, tumor-induced osteoclastogenesis was sensitive to CBD and resistant to denosumab, whereas RANKL-induced osteoclastogenesis was resistant to CBD and sensitive to denosumab. These results suggest that IL-1 secreted from OSCC cells stimulates not only tumor-induced osteoclastogenesis targeting OPCs but also physiological RANKL-induced osteoclastogenesis, and this may be the biological mechanism of bone resorption induced by the infiltration of OSCC. These results also suggest that IL-1 inhibitors are potential therapeutic agents against bone resorption induced by OSCC, leading to improved prognosis and quality of life among OSCC patients.

## Figures and Tables

**Figure 1 ijms-24-00688-f001:**
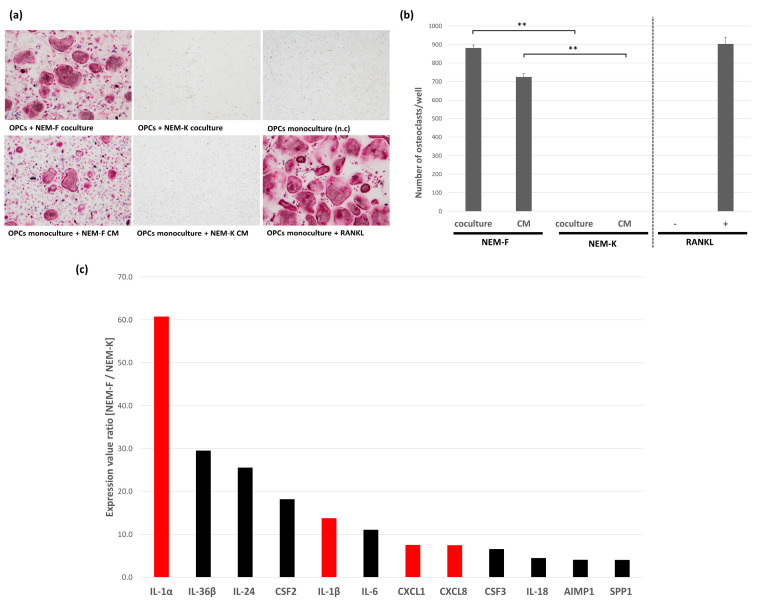
Osteoclastogenic properties of two subclones derived from the OSCC cell line, NEM. (**a**) Representative views of TRAP staining 5 days after the coculture of NEM-F or NEM-K cells with OPCs, and 5 days after the culture with culture supernatant of NEM-F or NEM-K cells. After 5 days of culturing OPCs in the growth medium supplemented with or without RANKL, the cells also functioned as positive and negative controls. (**b**) The number of osteoclasts induced by NEM-F relative to that induced by NEM-K, and the number of osteoclasts induced in cultures of OPCs supplemented with RANKL relative to that without supplementation of RANKL in the growth medium. ** *p* < 0.01 (one-way ANOVA with the Tukey–Kramer method). (**c**) Gene expression value ratio in NEM-F relative to that in NEM-K, analyzed with the gene expression microarray, sorted by the ratio as “>4-fold” and the GO Molecular Function Term as “cytokine activity”. Red bars represent selected inflammatory cytokines that have been reported to have an association with osteoclastogenesis. Preferential expression of the selected inflammatory cytokines was also detected in a gene expression microarray analysis used in our previous study (NCBI Gene Expression Omnibus, GE12341).

**Figure 2 ijms-24-00688-f002:**
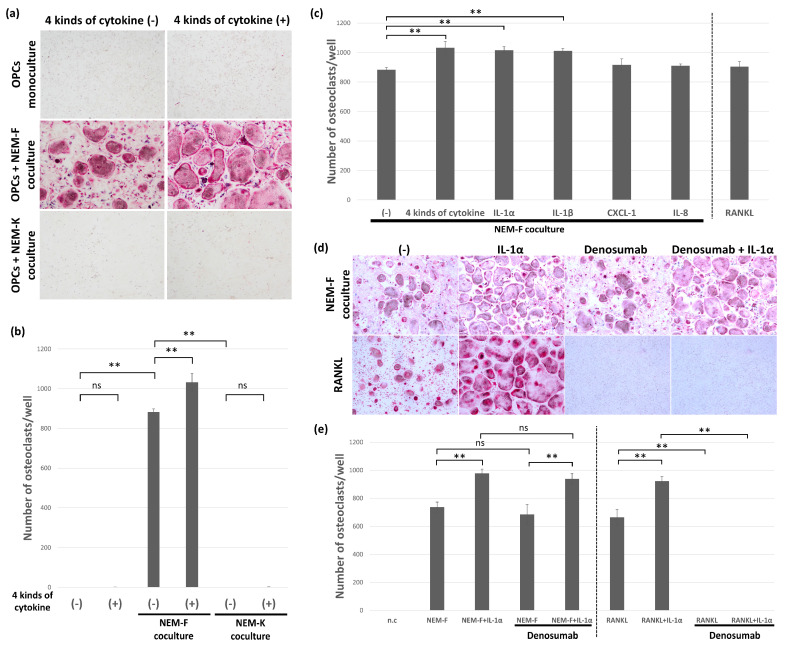
Characterization of selected inflammatory cytokines on stimulation or induction of osteoclastogenesis. (**a**) Representative views of TRAP staining 5 days after the coculture of NEM-F or NEM-K cells with OPCs with or without supplementation of a mixture of IL-1α, IL-1β, IL-8, and CXCL1 and 5 days after the culture of OPCs with or without supplementation of a mixture of IL-1α, IL-1β, IL-8, and CXCL1. (**b**) The number of osteoclasts 5 days after the coculture of NEM-F or NEM-K cells with OPCs with or without supplementation of a mixture of IL-1α, IL-1β, IL-8, and CXCL1 and 5 days after the culture of OPCs with or without supplementation of a mixture of IL-1α, IL-1β, IL-8, and CXCL1. ** *p* < 0.01, ns: not significant (one-way ANOVA with the Tukey–Kramer method). (**c**) Quantitative data on the number of osteoclasts 5 days after the coculture of NEM-F cells with OPCs supplemented with each inflammatory cytokine. ** *p* < 0.01 (one-way ANOVA with the Tukey–Kramer method). (**d**) Evaluation of the effects of denosumab on osteoclastogenesis induced by the coculture of NEM-F cells with OPCs or OPCs cultured with RANKL with or without supplementation of IL-1α. Representative views of TRAP staining after 5 days of coculture. (**e**) Quantitative data on the number of osteoclasts induced by the coculture of NEM-F cells with OPCs or OPCs cultured with RANKL with or without supplementation of IL-1α and further supplementation of denosumab. ** *p* < 0.01, ns: not significant (one-way ANOVA with the Tukey–Kramer method).

**Figure 3 ijms-24-00688-f003:**
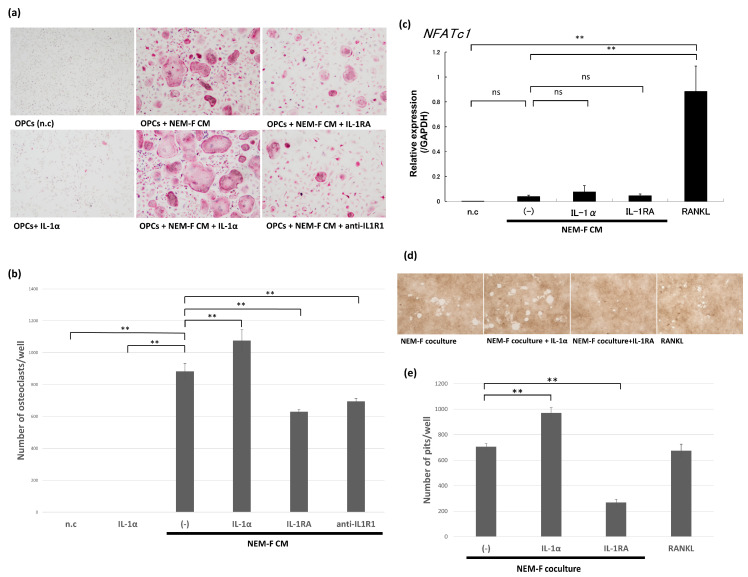
The effects of IL-1 and IL-1RA on osteoclastogenesis induced by the culture supernatant of NEM-F cells and on the expression of NFATc1 in these cultures. (**a**) Evaluation of the effects of IL-1, IL-1RA, and an antagonistic antibody for IL-1R1 on osteoclastogenesis induced by the culture supernatant of NEM-F cells. Representative views of TRAP staining after 5 days of culture. (**b**) The number of osteoclasts induced by the culture supernatant of NEM-F cells with or without treatment with IL-1, IL-1RA, or an antagonistic antibody for IL-1R1. ** *p* < 0.01 (one-way ANOVA with the Dunnett’s method). (**c**) Quantitative PCR analysis of the expression of NFATc1 in OPCs cultured with culture supernatant of NEM-F cells with or without supplementation of IL-1RA or IL-1α relative to the expression in OPCs cultured with the growth medium with or without supplementation of RANKL. ** *p* < 0.01, ns: not significant (one-way ANOVA with the Tukey–Kramer method). (**d**) Evaluation of the effects of IL-1 and IL-1RA on the osteolytic function of induced osteoclasts by the coculture of NEM-F cells and OPCs using a pit formation assay. Representative views of formatted pits after 6 days of culture. (**e**) The number of pits formed by osteoclasts induced by the coculture of NEM-F cells and OPCs with or without supplementation of IL-1RA or IL-1α. ** *p* < 0.01 (one-way ANOVA with the Dunnett’s method).

**Figure 4 ijms-24-00688-f004:**
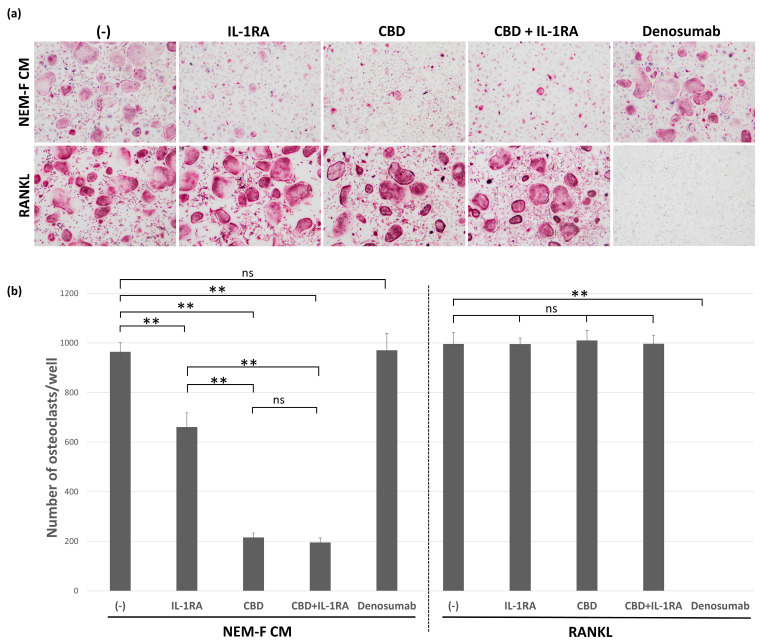
The effects of IL-1RA and CBD on osteoclastogenesis induced by the culture supernatant of NEM-F cells and RANKL. (**a**) Evaluation of IL-1RA, CBD, and denosumab on osteoclastogenesis induced by the culture supernatant of NEM-F cells and RANKL. Representative views of TRAP staining after 5 days of culture. (**b**) The number of osteoclasts induced by the culture supernatant of NEM-F cells and RANKL, with or without treatment of IL-1RA, CBD, and denosumab. ** *p* < 0.01, ns: not significant (one-way ANOVA with the Tukey–Kramer method).

**Figure 5 ijms-24-00688-f005:**
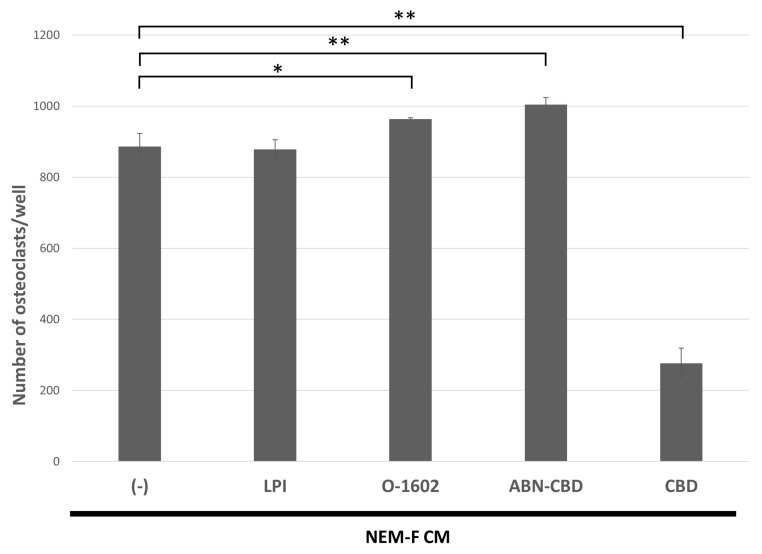
The effect of GPR55 antagonists or agonists on osteoclastogenesis induced by the culture supernatant of NEM-F cells. The number of osteoclasts induced by the culture supernatant of NEM-F cells supplemented with GPR55 antagonists, such as CBD, or agonists, such as LPI, O-1602, and ABN-CBD, relative to those without supplementation. ** *p* < 0.01, * *p* < 0.05 (one-way ANOVA with the Dunnett’s method).

**Figure 6 ijms-24-00688-f006:**
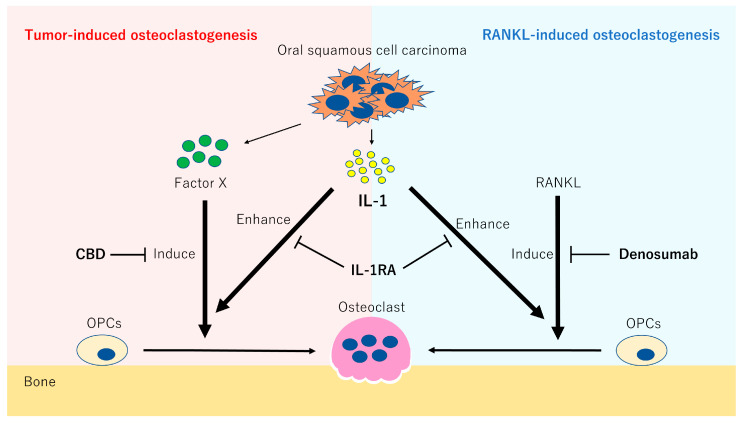
Hypothesis of the mechanism of bone resorption induced by the infiltration of OSCC.

## Data Availability

Not applicable.

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
