# Peer review of "IL-1 Generated by Oral Squamous Cell Carcinoma Stimulates Tumor-Induced and RANKL-Induced Osteoclastogenesis: A Possible Mechanism of Bone Resorption Induced by the Infiltration of Oral Squamous Cell Carcinoma"

_ijms, 2022, doi:10.3390/ijms24010688_

Round 1
Reviewer 1 Report
The paper is really interesting and it explores the relation between IL-1, oral squamous cell carcinoma and osteoclastogenesis. The study is well conducted and it proves a lot of results enriched by a good iconography and captions. The results are well discussed and they open interesting reflections of future applications of these results in adiuvant therapy. A minor suggestion would be to put "material and methods" as second section of the article, not the forth, for a greater knowledge of the settings. A question may be moved: what about p16 positive cancers? even though they are more frequent in the oropharynx, what about their molecular behaviour?
Author Response
Thank you very much for your favorable comments to our manuscript. As the reviewer suggested, materials and methods section was thought to be better at the second section rather than the forth section in this article. However, based on the editorial policy of this journal, we are unable to move it to the second section.
We did not mention p16 in this study. As the reviewer suggested, the frequency of p16-positivity in OSCC is much lower than that of oropharynx cancers. Current WHO Classification of Head and Neck Tumours (4th Edition, 2017) describes that “HPV, in particular type 16 is a recognized etiological factor in oropharyngeal cancer but is only seen in a small minority (3%) of OSCCs” (page 109). Hence, we did not mention p16 in this manuscript. Etiology of oral squamous cell carcinoma would be very different from that of oropharyngeal cancer.
Reviewer 2 Report
In the manuscript entitled “IL-1 generated by oral squamous cell carcinoma stimulates tumor-induced and RANKL-induced osteoclastogenesis: a possi
ble mechanism of bone resorption induced by infiltration of oral squamous cell carcinoma” (ijms-2058794), the authors were trying to clarify OSCC-induced osteoclastogenesis by IL-1 and Rankle secreted by oral squamous cell carcinoma (OSCC) cells. Oral squamous cell carcinoma inducing osteoclastogenesis were reported by several published manuscript. Also, it was dementated both IL-1 and Rankle secreted by different cancer cells or other resourses, including OSCC, induced osteoclastogenesis. Based on current published evidences, the current manuscript had no novelty at all.

Author Response
In the previous study, we discovered the novel tumor-induced osteoclastogenesis targeting osteoclast precursor cells (OPCs) rather than osteoclast progenitor cells. In this study, we clarified biological significance of IL-1 secreted from OSCC cells in tumor-induced and RANKL-induced osteoclastogenesis. We analyzed biological significance of IL-1 in tumor-induced osteoclastogenesis and clarified that IL-1 stimulated osteoclastogenesis induced by OSCC cells from OPCs without affecting the expression of NFATc1. IL-1 also stimulated RANKL-induced osteoclastogenesis. With or without IL-1 supplementation, tumor-induced osteoclastogenesis was sensitive to CBD and resistant to denosumab, whereas RANKL-induced osteoclastogenesis was resistant to CBD and sensitive to denosumab. They are novel findings in this field. These results suggest that IL-1 secreted from OSCC cells stimulates not only tumor-induced osteoclastogenesis targeting OPCs but physiological RANKL-induced osteoclastogenesis, and this may be the biological mechanism of bone resorption induced by infiltration of OSCC. These results also suggest that IL-1 inhibitors are candidates for therapeutic agents against bone resorption induced by OSCC.
The manuscript has been edited by MDPI English editing service.
Reviewer 3 Report
The paper is interesting and encompasses an important aspect. OSCC is a common entity and has devastating effects on sufferers. the co-morbidity like osteoclastic activity initiation is a problem and needs to be explored further.
Comments and suggestions are to be addressed in the paper before further processing.
Abstract: The authors need to include methods and results precisely in the abstract section. Reduce the background information. the conclusion is there but not consistent with the main section.
Introduction: Include the information "what is still not covered or loopholes in the scientific literature yet not available. mention the novelty of your study if any.
moreover, the justification of doing this study also needs elaboration, as in how this study outcome will benefit the clinicians or OSCC sufferers.
The methods and result section is well explained.
Discussion: add limitations, strengths, and future recommendations in the last 2 paragraphs about this study.
Conclusion: the heading is missing, mention the main achievements as per the objective of the study.
Author Response
The paper is interesting and encompasses an important aspect. OSCC is a common entity and has devastating effects on sufferers. the co-morbidity like osteoclastic activity initiation is a problem and needs to be explored further.
Comments and suggestions are to be addressed in the paper before further processing.
Abstract: The authors need to include methods and results precisely in the abstract section. Reduce the background information. the conclusion is there but not consistent with the main section.
We appreciate the reviewer’s advice. Following the reviewer’s comments, we added methods in abstract and described aims and conclusions of this study more clearly.
Abstract was changed from “We previously observed a novel osteoclastogenesis system that is induced by oral squamous cell carcinoma (OSCC) cells, which target osteoclast precursor cells without upregulation of the master transcriptional factor of osteoclastogenesis, NFATc1. OSCC-induced osteoclastogenesis was effectively attenuated by cannabidiol but denosumab was ineffective. Here, we analyzed inflammatory cytokines that were preferentially expressed in one of the osteoclastogenic OSCC cell lines, namely NEM, compared with the subclone that had lost the osteoclastogenic property. Among IL-1, IL-6, IL-8, and CXCL1, which we chose as candidates responsible for tumor-induced osteoclastogenesis, IL-1 stimulated osteoclastogenesis induced by NEM cells, although IL-1 alone did not reveal osteoclastogenesis. Stimulation of RANKL-induced osteoclastogenesis by IL-1 was also confirmed. The IL-1 receptor antagonist significantly attenuated osteoclastogenesis induced by NEM cells. The stimulatory effects of IL-1 for OSCC-induced and RANKL-induced osteoclastogenesis were effectively attenuated with cannabidiol and denosumab, respectively. These results suggest that IL-1 secreted from OSCC cells stimulates not only tumor-induced osteoclastogenesis targeting OPCs but physiological RANKL-induced osteoclastogenesis, and this may be the mechanism of bone resorption induced by infiltration of OSCC. IL-1 inhibitors may be candidates for therapeutic agents against bone resorption induced by OSCC.” To “We previously observed a novel osteoclastogenesis system that is induced by oral squamous cell carcinoma (OSCC) cells, which target osteoclast precursor cells (OPC) without upregulation of the master transcriptional factor of osteoclastogenesis, NFATc1. Here, we analyzed inflammatory cytokines that were preferentially expressed in one of the osteoclastogenic OSCC cell lines, namely NEM, compared with the subclone that had lost its osteoclastogenic properties. Based on a gene expression microarray and a protein array analyses, IL-1, IL-6, IL-8, and CXCL1 were chosen as candidates responsible for tumor-induced osteoclastogenesis. From the results of the in vitro osteoclastogenesis assay using OPCs cultured with OSCC cells or their culture supernatants, IL-1 was selected as a stimulator of both OSCC-induced and RANKL-induced osteoclastogenesis. The IL-1 receptor antagonist significantly attenuated osteoclastogenesis induced by NEM cells. The stimulatory effects of IL-1 for OSCC-induced and RANKL-induced osteoclastogenesis were effectively attenuated with cannabidiol and denosumab, respectively. These results suggest that IL-1 secreted from OSCC cells stimulates not only tumor-induced osteoclastogenesis targeting OPCs but also physiological RANKL-induced osteoclastogenesis, and this may be the biological mechanism of bone resorption induced by the infiltration of OSCC. These results also suggest that IL-1 inhibitors are candidates for therapeutic agents against bone resorption induced by OSCC.”. Please refer to abstract in the revised version.
Introduction: Include the information "what is still not covered or loopholes in the scientific literature yet not available. mention the novelty of your study if any.
moreover, the justification of doing this study also needs elaboration, as in how this study outcome will benefit the clinicians or OSCC sufferers.
We appreciate the reviewer’s appropriate comments. Following the reviewer’s comments, we extensively rewrote introduction section to clarify loopholes in this field and to show the aim and values of this study more clearly. Please refer to introduction in the revised version.
The methods and result section is well explained.
Discussion: add limitations, strengths, and future recommendations in the last 2 paragraphs about this study.
Following the reviewer’s valuable comments, the last paragraph of discussion was changed from “Moreover, IL-1 is expressed in head and neck carcinomas. In the current study, we showed that IL-1RA, an inhibitor for IL-1α and IL-1β, significantly attenuated osteoclastogenesis induced by NEM-F cells. These results suggest that IL-1 generated by OSCC cells participated in tumor-induced osteoclastogenesis. In addition, we confirmed the stimulatory effect of IL-1α on RANKL-induced osteoclastogenesis. Overall, IL-1 secreted from OSCC cells may spread by diffusion, act on OPCs and osteoclast progenitor cells near the bone surface and stimulate not only tumor-induced osteoclastogenesis targeting OPCs but physiological RANKL-induced osteoclastogenesis on the bone surface. Hence, IL-1 inhibitors may effectively attenuate bone resorption caused by OSCC infiltration by suppressing the stimulatory effect of IL-1 on tumor-induced and conventional RANKL-induced osteoclastogenesis (Fig. 6). In addition, IL-1RA, which has been applied clinically for the treatment of rheumatoid arthritis in some countries, may be one of the potential therapeutic agents against OSCC-induced bone resorption.” (page 10, lines 313 to 325 in the original version) to “Moreover, IL-1 is expressed in head and neck carcinomas [24, 25, 33]. In the current study, we showed that IL-1RA, an inhibitor for IL-1α and IL-1β, significantly attenuated osteoclastogenesis induced by NEM-F cells. These results suggest that IL-1 generated by OSCC cells participated in tumor-induced osteoclastogenesis. In addition, we confirmed the stimulatory effect of IL-1α on RANKL-induced osteoclastogenesis. Overall, IL-1 secreted from OSCC cells may spread by diffusion, act on OPCs and osteoclast progenitor cells near the bone surface and stimulate not only tumor-induced osteoclastogenesis targeting OPCs but also physiological RANKL-induced osteoclastogenesis on the bone surface. Hence, IL-1 inhibitors may effectively attenuate bone resorption caused by OSCC infiltration by suppressing the stimulatory effect of IL-1 on tumor-induced and conventional RANKL-induced osteoclastogenesis (Fig. 6). In this study, there are no in vivo data, and further animal experiments are needed to confirm our hypothesis. In addition, IL-1RA, which has been applied clinically for the treatment of rheumatoid arthritis in some countries, may be one of the potential therapeutic agents against OSCC-induced bone resorption. Considering that most of OSCC cell lines possess osteoclastogenic properties from OPCs, the results of this study provide valuable information for developing new therapeutic agents against bone resorption induced by OSCC, contributing to the improvement of prognoses and quality of life among OSCC patients.” (page 10, lines 326 to 343 in the revised version). Please refer to the revised version.
Conclusion: the heading is missing, mention the main achievements as per the objective of the study.
Following the reviewer’s advice, we added the heading and changed conclusions to show the objective of this study more clearly. We changed conclusion from “In conclusion, we clarified that IL-1 stimulated osteoclastogenesis induced by OSCC cells from OPCs without affecting the expression of NFATc1. IL-1 also stimulated RANKL-induced osteoclastogenesis. With or without IL-1 supplementation, tumor-induced osteoclastogenesis was sensitive to CBD and resistant to denosumab, whereas RANKL-induced osteoclastogenesis was resistant to CBD and sensitive to denosumab. IL-1 secreted from OSCC cells may stimulate not only tumor-induced osteoclastogenesis targeting OPCs but physiological RANKL-induced osteoclastogenesis, and IL-1 inhibitors may be potential therapeutic agents against bone resorption induced by OSCC.” (page 10, lines 326 to 333 in the original version) to “5. Conclusions We analyzed the biological significance of IL-1 in tumor-induced osteoclastogenesis and clarified that IL-1 stimulated osteoclastogenesis induced by OSCC cells from OPCs without affecting the expression of NFATc1. IL-1 also stimulated RANKL-induced osteoclastogenesis. With or without IL-1 supplementation, tumor-induced osteoclastogenesis was sensitive to CBD and resistant to denosumab, whereas RANKL-induced osteoclastogenesis was resistant to CBD and sensitive to denosumab. These results suggest that IL-1 secreted from OSCC cells stimulates not only tumor-induced osteoclastogenesis targeting OPCs but physiological RANKL-induced osteoclastogenesis, and this may be the biological mechanism of bone resorption induced by the infiltration of OSCC. These results also suggest that IL-1 inhibitors are potential therapeutic agents against bone resorption induced by OSCC, leading to improved prognosis and quality of life among OSCC patients.” (page 12, lines 445 to 457 in the revised version). Please refer to the revised version.